# Enhanced Antioxidant and Neuroprotective Properties of Pterostilbene (Resveratrol Derivative) in Amorphous Solid Dispersions

**DOI:** 10.3390/ijms25052774

**Published:** 2024-02-28

**Authors:** Natalia Rosiak, Ewa Tykarska, Judyta Cielecka-Piontek

**Affiliations:** 1Department of Pharmacognosy and Biomaterials, Faculty of Pharmacy, Poznan University of Medical Sciences, 3 Rokietnicka St., 60-806 Poznan, Poland; nrosiak@ump.edu.pl; 2Department of Chemical Technology of Drugs, Poznan University of Medical Sciences, 3 Rokietnicka St., 60-806 Poznan, Poland; etykarsk@ump.edu.pl

**Keywords:** pterostilbene, polyvinylpyrrolidone, amorphous solid dispersion, glass transition, Gordon–Taylor equation, Couchman–Karasz equation, molecular modeling, miscibility, neuroprotection, antioxidation

## Abstract

In this study, amorphous solid dispersions (ASDs) of pterostilbene (PTR) with polyvinylpyrrolidone polymers (PVP K30 and VA64) were prepared through milling, affirming the amorphous dispersion of PTR via X-ray powder diffraction (XRPD) and differential scanning calorimetry (DSC). Subsequent analysis of DSC thermograms, augmented using mathematical equations such as the Gordon–Taylor and Couchman–Karasz equations, facilitated the determination of predicted values for glass transition (T_g_), PTR’s miscibility with PVP, and the strength of PTR’s interaction with the polymers. Fourier-transform infrared (FTIR) analysis validated interactions maintaining PTR’s amorphous state and identified involved functional groups, namely, the 4′–OH and/or –CH groups of PTR and the C=O group of PVP. The study culminated in evaluating the impact of amorphization on water solubility, the release profile in pH 6.8, and in vitro permeability (PAMPA-GIT and BBB methods). In addition, it was determined how improving water solubility affects the increase in antioxidant (ABTS, DPPH, CUPRAC, and FRAP assays) and neuroprotective (inhibition of cholinesterases: AChE and BChE) properties. The apparent solubility of the pure PTR was ~4.0 µg·mL^−1^ and showed no activity in the considered assays. For obtained ASDs (PTR-PVP30/PTR-PVPVA64, respectively) improvements in apparent solubility (410.8 and 383.2 µg·mL^−1^), release profile, permeability, antioxidant properties (ABTS: IC_50_ = 52.37/52.99 μg·mL^−1^, DPPH: IC_50_ = 163.43/173.96 μg·mL^−1^, CUPRAC: IC_0.5_ = 122.27/129.59 μg·mL^−1^, FRAP: IC_0.5_ = 95.69/98.57 μg·mL^−1^), and neuroprotective effects (AChE: 39.1%/36.2%, BChE: 76.9%/73.2%) were confirmed.

## 1. Introduction

One of the well-known stilbenes widely recognized for its properties is resveratrol (RSV). However, pterostilbene (PTR, 3,5-di-methoxy resveratrol derivative), discovered later, is emerging as a compound that surpasses resveratrol in certain aspects. Due to their similar structures, both compounds have similar properties [1]. However, the difference in the number of hydroxyl groups contributes to greater lipophilicity, better bioavailability, and increased biological activity of PTR compared to RSV [2,3,4]. PTR exhibits resistance to metabolic modifications, allowing a higher fraction of ingested PTR to reach its target sites and remain unaltered for a longer time compared to RSV [1,5,6]. For this reason, PTR is considered a more efficient and promising compound in various formulations compared to RSV.

PTR has recently attracted significant interest due to its potential neuroprotective, antioxidant, anti-inflammatory, anti-type 2 diabetes, anti-apoptotic, and anti-aging effects [7,8,9,10,11,12,13,14,15]. PTR is mostly found in blueberries and *Pterocarpus marsupium* wood trees [16,17,18,19]. PTR was determined to be pharmacologically safe. According to Obrador et al., it exhibited no organ-specific or systemic toxicity even when administered intravenously at a high dose (30 mg/kg per day for 23 days) [20]. Based on the available data, orally administered PTR appears safe at a dose of 125 mg administered twice daily (NCT01267227) [21,22].

Nonetheless, the limited solubility of PTR poses a challenge to its potential as a medicinal agent, potentially obstructing its delivery to specific tissues, such as the brain. To address this challenge, researchers have explored various methods. These approaches include the development of amorphous solid dispersion [23], inclusion complex [24,25], nanoparticles [26], cocrystals [27,28,29], and the production of nanoemulsions [30].

For example, PTR complexed with HPβCD exhibits potent antibacterial effects against *Fusobacterium nucleatum*, a key periodontal pathogen. Additionally, its antimicrobial and immunomodulatory properties make PTR a promising candidate for nutraceutical use in the adjunctive treatment of periodontitis [24,25]. Nanoparticles of PTR obtained by Zhao et al. successfully decreased blood glucose levels, enhanced antioxidant capacity, and reversed micro-inflammation in mice [26]. Bofill et al. confirmed that after oral administration of PTR:picolinic acid cocrystal (1:1) to rats, the relative bioavailability of PTR increased significantly (9.9 times compared to its commercially available solid oral form) [29].

PTR has been investigated in a number of preclinical [8,9,11,12,31,32,33] and clinical [21,22,34] studies. In preclinical investigations, PTR has shown potential as a hepatoprotective agent in cases of acute liver failure induced by various pharmaceutical drugs [35,36]. Shi et al. presented findings from a preclinical study indicating that PTR demonstrates hepatoprotective effects in instances of hepatic ischemia/reperfusion injury [31]. Furthermore, PTR may mitigate atherosclerosis induced by high-fat diets by suppressing several proinflammatory cytokines, as evidenced in a mice model [9].

The neuroprotective potential of PTR is also described in the literature. Animal models that simulate Alzheimer’s disease and cognition have been used to study the in vivo effects of PTR [37]. In a dose-dependent manner, it was shown that adding PTR to one’s diet was beneficial in reversing the negative effects of aging on cognitive function, particularly working memory. Another research study investigated the neuroprotective effects of PTR in treating memory damage in Sprague–Dawley rats when streptozotocin was administered intracerebroventricularly [38]. PTR mitigated the memory loss caused by streptozotocin. Additionally, after consuming PTR, some anti-oxidant parameters improved, and cholinergic neurotransmission was enhanced, as shown by a drop in acetylcholinesterase (AChE) activity. Chang et al. confirmed that therapy with PTR at a dose of 120 mg/kg per week for eight weeks can relieve significant pathological changes in a mouse model of Alzheimer’s disease [11].

In addition, the in vivo investigation of an Alzheimer’s disease model, conducted on senescence-accelerated mice, affirmed the neuroprotective impact of PTR [11]. Another study demonstrated that PTR pretreatment successfully reversed induced learning and memory impairment in mice [12]. Numerous studies confirmed PTR’s antioxidant properties through both in vitro and in vivo experiments, demonstrating its preventive and therapeutic potential. PTR’s antioxidant capabilities have been associated with its role in preventing cancer, influencing neurological conditions, reducing inflammation, mitigating vascular diseases, and improving diabetes outcomes [2,21,39]. Moreover, the in silico study of PTR activity has been of interest in various biomedical applications [40,41,42,43,44,45]. PTR has been investigated for, among other reasons, (i) the binding mode with the Keap1 Kelch domain [41], (ii) a telomerase inhibitor [43], and (iii) to confirm whether PTR binds to the MMP-9 proteins [42]. The latest literature reports strengthen the validity of combining theoretical methods with in vitro methods. For instance, Rahmani et al. [46] validated, through molecular modeling, the interaction between the primary components of myrrh extract and the functional residues of the SARS-CoV-2 virus’s spike protein. Studzińska-Sroka et al. [47] demonstrated through in vitro and in silico investigations that caperatic acid (the principal compound in *Platismatia glauca* extract) influences the extract’s potential against cholinesterases. They confirmed that the effect of the extract against butyrylcholinesterase (BChE) is stronger than against AChE. In another study [48], both experimental and theoretical analyses verified that atranorin possesses a high capacity to inhibit AChE. Alhaithloul et al. [49] utilized molecular docking to show that limonene (the main compound in the rosemary methanolic extract) forms stable complexes with DNA gyrase A.

Amorphous solid dispersions (ASDs) enhance the solubility, dissolution rate, and bioavailability of polyphenols and other poorly water-soluble drugs. They are prepared by dispersing the active substance in a carrier, usually a pharmacologically inert polymer. Techniques such as hot melt extrusion [50], solvent evaporation [51], ball milling [52], spray drying [53], supercritical carbon dioxide extraction [54], and freeze drying [55] are used to obtain the ASDs of polyphenols. Due to the thermodynamic instability of amorphous drugs, their recrystallization may occur during storage. Therefore, the challenge is to select the appropriate polymer and its proportions in the solid dispersion to prevent the potential recrystallization of the dispersion.

In the present study, an attempt was made to obtain ASDs of PTR using polyvinylpyrrolidone (PVP) polymers, aiming to enhance physical stability and water solubility compared to previously published PTR-Soluplus^®^ ASDs [23]. We also examined the antioxidant and neuroprotective properties of prepared PTR ASDs. It is the first time we investigated the potential of PTR to inhibit AChE and BChE enzymes in both experimental and in silico studies.

## 2. Results and Discussion

In the present study, amorphous solid dispersions (ASDs) of pterostilbene (PTR) with polyvinylpyrrolidone polymers (PVP K30 and VA64) were prepared using milling. The amorphous nature of PTR was confirmed by X-ray powder diffraction (XRPD) and differential scanning calorimetry (DSC). The analysis of DSC thermograms, complemented with the use of Gordon–Taylor and Couchman–Karasz mathematical equations, allowed the determination of experimental and predicted values for the glass transition (T_g_), the miscibility of PTR with PVPs, and the strength of PTR’s interaction with the polymers. Fourier-transform infrared (FTIR) analysis was employed to validate the interactions responsible for maintaining PTR’s amorphous state and identify the functional groups. The final stage of the study involved assessing the impact of amorphization on water solubility, release profile, permeability, and biological activity, specifically conducting antioxidant and neuroprotective activity.

Compared to conventional methods that include the use of organic solvents or high-energy procedures that produce a lot of waste [56,57,58,59], ball milling offers a “green” way to manufacture ASDs. In the literature, ball milling was used to create ASDs containing polyphenols [23,60,61,62,63] or active pharmaceutical ingredients (APIs) [64,65,66,67].

A commonly employed technique to discern the crystalline or amorphous nature of a sample is XRPD analysis. Hence, in our studies, this method was used to confirm the crystallinity of pure PTR and PTR in physical mixtures, as well as to verify the amorphous nature of PTR-PVP after introduction to ASDs.

Pure PTR’s powder diffraction patterns revealed recognizable, strong Bragg’s peaks, which confirmed its crystalline structure (Figure 1a,b, black line). This aligns with information found in the literature [23,27,68]. Due to the amorphous nature of PVP30 and PVPVA64, no crystalline peaks were observed in the diffractograms (Figure 1a,b, red line). According to the literature, the amorphous form of PVP K30 and PVP VA64 exhibits two broad non-Bragg peaks with a maximum at about 12° 2Θ and 20° 2Θ [69].

The diffractograms of physical mixtures and solid dispersions were then compared to the XRPD patterns of PTR and polymers. The PTR and PVP peaks were observed in the diffractograms of each physical mixture (Figure 1, blue line). Our earlier research verified that physical mixtures of polyphenol and polymer exhibit distinct polyphenol peaks with reduced intensity [23,60,70]. The disappearance of Bragg’s peaks in the diffractograms of PTR-PVP30 ASD and PTR-PVPVA64 ASD (Figure 1, green line) is an indicative sign of the loss of long-range order associated with crystalline structures. The lack of Bragg’s peaks (effect “halo”) shows that PTR is present in the PVP matrix in an amorphous state. This result is consistent with other studies that demonstrated PVP30 and PVPVA64′s capacity to encourage the amorphous state of polyphenol molecules [71,72,73,74].

According to the results of the thermogravimetric (TG) study, PTR is stable up to ~200 °C, whereas PVP K30 and PVP VA64 are stable to 365 °C and 270 °C, respectively (Figure 2). In this range, the TG of polymers shows a loss of mass of ~5% for PVP30 and ~1% for PVPVA64 due to the evaporation of moisture. The data that were gathered made it possible to choose the right temperature range for the DSC study.

The amorphousness of PTR in ASD containing PVP and the miscibility of the resulting ASD were both confirmed in our work using DSC analysis. A melting peak of the crystalline phase of PTR was detected at a temperature of roughly 98.2 °C (Figure 2, black dashed line). The findings agree with data from the literature that suggests PTR has a melting point (T_m_) of 100 °C [23,75,76]. Lacerda et al. [75] and Catenacci et al. [76] observed T_m_ of PTR at 94.55 °C and 95.9 °C, respectively.

The PTR-PVP ASD thermograms did not exhibit any endothermic peaks (see Figure 3) that point to the presence of a crystalline phase in the sample. Additionally, perfect miscibility is shown by the observation of a single glass transition temperature (T_g_) in the ASDs (PTR-PVP30: 124.6 °C, PTR-PVPVA64: 93.8 °C). The results obtained indicate pterostilbene’s amorphous nature in these ASDs.

The T_g_ of an ASD is reflective of the interactions between the two components. The relationship between the T_g_ changes in ASD and the molecular-level interactions helps us understand the mathematical model of Gordon−Taylor (G–T) and Couchman–Karasz (C–K) equations (Equation (1)) [23]:(1)$$T_{g}=\frac{w_{1}T_{g1}+kw_{2}T_{g2}}{w_{1}+kw_{2}}$$
where T_g_ is the glass transition temperature of a mixture with weight fractions ω_1_ and ω_2_ of PTR and PVP, whose glass transition temperatures are T_g1_ and T_g2_, respectively. k for G-T is a parameter that can be estimated using the Simha–Boyer equation (k = ($\rho_{1}$T_g1_)·($\rho_{2}$T_g2_)^−1^), where $\rho_{1},\rho_{2}$—the densities of PTR and PVP, respectively, whereas k for C-K is k = ${∆c}_{p2}$·${∆c}_{p1}$^−1^, where ∆c_p1_ and ∆c_p2_ are the change in the heat capacity at T_g1_ and T_g2_, respectively.

In our previous study [23], the ball milling process failed to obtain amorphous pterostilbene (PTR). Consequently, the Tg of PTR was ascertained through the melting and cooling method within a DSC oven, and this value was employed in the G-T and C-K equations (XRPD analysis confirmed the amorphous state of melted PTR). The same approach was used by Löbmann et al. for carbamazepine [77], Rosiak et al. for hesperidin and kaempferol [52,60], Wdowiak et al. for curcumin, hesperetin, and piperine [50,70], and Garbiec et al. for genistein [78].

The T_g_ values observed for PTR-PVP ASDs were compared to the T_g_ predicted using the G-T and C-K equations and summarized in Table 1.

In our study, PVP was intimately mixed with PTR (component with a lower T_g_) in ASDs (observed one T_g_). This suggests that PTR was ‘‘dissolved’’ in the amorphous material (PVP) and enhanced its molecular mobility by increasing the spacing between polymer chains and free volume. As a result, a decrease in the T_g_ of ASDs was observed (this process is known as plasticization) [79], whereby the T_g_ of the obtained ASDs was between the T_g_ of PTR and T_g_ of PVP. The literature suggests that a homogeneous two-component mixture’s T_g_ typically lies between the T_g_s of each component [77,80]. In addition, the positive deviation (T_g,exp_ > T_g,G-T or C-K_) of the experimental T_g_ of PTR-PVP30 and PTR-PVPVA64 compared to its predicted T_g_ from the G-T and C-K equations suggests specific molecular interactions between PTR and PVP in ASDs [77,81]. According to the literature, a positive deviation might indicate a potent hetero-nuclear interaction (such as H-bonding, etc.) [23,82]. This demonstrates the ASD’s capacity to forge strong bonds, which raises the T_g,exp_ value.

Several investigators have employed FTIR spectroscopy to analyze the interactions accountable for the formation of ASD [23,50,60,70,83,84,85,86,87,88]. The examination of FTIR spectra corresponding to the different bonds sheds light on intermolecular interactions. When electronegative atoms (such as oxygen or nitrogen) and hydrogen atoms come together, hydrogen bonds are formed. The vibrational frequencies of the relevant functional groups are impacted by hydrogen bonding in ASD. Particularly, the stretching frequencies of bonds involved in hydrogen bonding, such as O-H or N-H, tend to shift. The hydrogen bonding-induced bond weakening, which changes the force constant and lowers the bond’s mass, is the cause of this shift. This shift is often accompanied by an increase in peak intensity and breadth due to the presence of multiple interactions [23,50,60,70,83,84,85,86,87,88]. Therefore, in our work, FTIR-ATR analysis in the MIR region (400–4000 cm^−1^) was used to identify the interactions responsible for maintaining the amorphous state of PTR in dispersions with PVP.

The chemical structure of PTR, PVP K30, and PVP VA64 are shown in Figure 4.

Infrared spectra of crystalline PTR bands were seen at around 400–1600 cm^−1^ (Figure 5a,b, black line) and 2800–3400 cm^−1^ (Appendix A, black line).

The –CH wagging, torsion, or bending vibrations are represented as dominating bands in the first range. Moreover, the –OC stretching was discernible at around 1050–1600 cm^−1^ (1049 cm^−1^, 1061 cm^−1^, 1319 cm^−1^, 1514 cm^−1^, and 1584 cm^−1^); in the same range, C–C–C stretching/asymmetric stretching was observed at 1049 cm^−1^, 1061 cm^−1^, 1319 cm^−1^, and 1514 cm^−1^. Predominant bands in the 2850–3005 cm^−1^ range correlate to the –CH_2_ or –CH_3_ stretching. At 3028 cm^−1^, CH stretching at the phenolic ring was evident. The –OH stretching has a sharp peak at around 3341 cm^−1^ [23].

PVP K30 is an amphiphilic and nonionic homopolymer of N-vinyl-pyrrolidone that contains a carbonic backbone with pyrrolidine rings attached [72]. Meanwhile, PVP VA64 is a vinyl-pyrrolidone-vinyl acetate copolymer [89]. A carbonyl (C=O) group found in the structure of PVP polymers allows them to form hydrogen bonds with pharmaceuticals that include proton donors and stabilize amorphous drug molecules [90,91].

FTIR spectra of pure PVP K30/PVP VA64 (Figure 5a and Figure 5b, red line, respectively) indicated peaks at 1167/- cm^−1^ (C–C=O) [92], 1229/1233 cm^−1^ (lactone structure) [93], 1285/1287 cm^−1^ (C–N stretching) [94], 1373/1369 cm^−1^ (–CH deformation modes from –CH_2_) [87], 1420/1422 cm^−1^ (CH_2_ wagging) [92], 1460/1460 cm^−1^ (CH_2_ bending) [95], 1665/1670 cm^−1^ (C=O absorption peak from amide group) [94], -/1732 cm^−1^ (C=O stretching of vinyl acetate) [96], and 2800–3100 cm^−1^ (C–H stretching) [94].

In the ASD spectrum, we observe the disappearance of numerous PTR bands, indicating its dispersion in the polymer matrix. In addition, the disappearance of the band corresponding to the 4′–OH stretching vibration (3341 cm^−1^ and 3370 cm^−1^) and the decrease in intensity and/or shift of the PTR bands observed at 1049 cm^−1^, 1061 cm^−1^, 1103 cm^−1^, 1146 cm^−1^, 1171 cm^−1^, 1200 cm^−1^, 1514 cm^−1^, 1584 cm^−1^ (predominant CH bond vibrations) indicates the involvement of these groups in hydrogen bond formation with PVP in all ASDs. In addition, the characteristic PVP30′s peak observed at 1665 cm^−1^ in the PTR-PVP30 occurs at about 1651 cm^−1^. However, the PVPVA64 peak at 1670 cm^−1^ shifted in PTR-PVPVA64 to 1659 cm^−1^. According to the literature [96], PVP30 possesses one hydrogen bond acceptor group, whereas PVP VA64 possesses two, which are derived from the carbonyl group of the pyrrolidone ring (at 1665 cm^−1^: PVP30 and 1670 cm^−1^: PVPVA64) and vinyl acetate (at 1732 cm^−1^: PVP VA64). The band shifts are indicative of the involvement of the C=O group in the formation of the hydrogen bond with the PTR.

FTIR investigations showed that the formation of hydrogen bonds between the 4′–OH and/or –CH groups of PTR and the C=O group of PVP made it feasible to create miscible amorphous PTR-PVP solid dispersion (confirmed by DSC). This is consistent with other research that found that crystallization or the formation of intermolecular hydrogen bonds frequently causes vibration peak disappearance and/or peak position shifts [97,98]. The formation of intermolecular hydrogen bonds was confirmed in our previous study for pterostilbene-Soluplus amorphous solid dispersion through the disappearance of many characteristic PTR peaks in the ASD and changes in characteristic Soluplus bands (shifting, decreased intensity, and/or disappearance of peaks) [23].

In addition, the literature confirms that the carbonyl groups present in PVP form hydrogen bonds with the hydroxyl groups of polyphenols. For example, de Mello Costa et al. [86] suggest the formation of hydrogen bonding between the carbonyl groups of PVP K25 (at the pyrrolidone ring) and the phenols aromatic group of quercetin. Gayo et al. [87] confirmed that the phenolic group of quercetin is able to form hydrogen bonds with the C=O group of PVP. These bonds allow quercetin to be dispersed in the PVP matrix and to remain in an amorphous form. According to research conducted by He et al. [72], curcumin and PVP K30 had an intermolecular hydrogen connection, and the carbonyl group of PVP’s pyrrolidone was a proton acceptor that formed a hydrogen bond with curcumin.

The main difficulty in formulating amorphous substances arises from their inherent instability, as they tend to shift from an energetically unfavorable amorphous state to a more stable crystalline structure. To tackle this challenge, a thorough investigation into the stability of the developed amorphous system was meticulously carried out, examining its performance during storage under stress conditions. Studies of physical stability for ASD provide knowledge about the lifetime of an amorphous state, which is associated with enhanced physicochemical qualities. The literature shows that temperature, humidity, and the polymer carrier are factors that affect recrystallization [99]. In addition, it has been confirmed that systems exposed to moisture tend to exhibit amorphous–amorphous phase separation [100] or recrystallization [60]. Rumondor et al. [101] confirmed that moisture disrupts drug–polymer interactions in ASDs obtained for hydrophobic drugs and leads to phase separation.

In our tests, under 40 °C/RH = 75% conditions, ASDs powders changed their form and resembled rubber. Similar results were observed by Fitzpatrick et al. [102] for pure PVP samples. In their study, exposure to elevated temperature and humidity resulted in a change in the PVP from a glassy to a rubbery state. For this reason, stability studies were carried out at the point of 0, 1, 2, and 3 months (XRPD analysis) only at 30 °C/RH = 65% (Appendix A). Over time, there is an observed shift in the positions of the two characteristic maxima of the PTR-PVP ASD. Nevertheless, the results of this analysis unveiled a noteworthy outcome—the amorphous system maintained stability and retained its entirely amorphous nature throughout the storage duration (three months). Observed changes in peak position may be related to water sorption by samples. Teng et al. [79] have demonstrated a strong correlation between the water content in PVP samples exposed to moisture and shifts in the position of the “halo” effect in XRPD patterns. To prevent the sample from water sorption, it would be necessary to keep the sample in an additional protective package.

HPLC analysis confirmed that the apparent solubility of the pure PTR was ~4.0 µg·mL^−1^. The apparent solubility of PTR increased as PTR was dispersed in the PVP matrix. Amorphization improved the apparent solubility by ~103-fold and ~96-fold for the PTR-PVP30 (410.8 ± 1.3 µg·mL^−1^) and PTR-PVPVA64 (383.2 ± 1.0 µg·mL^−1^) ASDs, respectively. Furthermore, the apparent solubility of the PTR-PVP ASD was higher than that obtained for the PTR-Soluplus (PTR-SOL) amorphous dispersions presented in our previous work (~37-fold and ~28-fold for PTR-SOL 1:2 *w*/*w* and PTR-SOL 1:5 *w*/*w*) [23].

For the PTR ASDs, the release rate of the PTR at pH 6.8 was significantly improved compared to the pure compound; however, the “spring and parachute” effect was observed (see Figure 6).

This effect involves the generation of a rapidly dissolving and supersaturating “spring” followed by the inhibition of precipitation, acting as a “parachute” to maintain the supersaturation state. For PTR-PVP30 ASD, the maximum PTR content was observed at 60 min (~666 μg·mL^−1^), while, for PTR-PVPVA, at the same time, the total amount was ~290 μg·mL^−1^. Then, a parachute effect was noted, in which the total amount of the compound decreased to approximately 581 μg·mL^−1^ and 188 μg·mL^−1^ for PTR-PVP30 and PTR-PVPVA, respectively. The plateau was observed from 1 h until the end of the study (7 h). Interestingly, this level was also confirmed after 24 h. For the pure compound, the maximum concentration was ~0.41 μg·mL^−1^ after 30 min, and a plateau was observed until the end of the study.

Knowing the significant impact of ASDs on the dissolution rate and solubility, their potential impact on the permeability of PTR was examined. The PAMPA models were carried out to simulate passive diffusion through gastrointestinal walls (GIT model) and the blood–brain barrier (BBB model) to determine whether the formulated ASDs enhanced the permeability of PTR in comparison to the pure compound.

Regarding the results obtained in PAMPA-GIT (see Appendix A), the P_app_ value of PTR was less than 0.1 × 10^−6^ cm·s^−1^ and suggests the low permeability of the pure compound, whereas, for the PTR-PVP ASDs, an increase in P_app_ value was observed. Improving the permeability of PTRs after the amorphization process classified ASDs in the medium permeability category. In the case of PAMPA-BBB (see Appendix A) for the PTR and PTR-PVP ASDs, the P_app_ value was kept above 1 × 10^−6^ cm·s^−1^, indicating that PTR maintained its high permeability (Appendix A). In addition, the results clearly show that amorphization significantly improved the permeability of PTR also in this model. The results confirmed that ASDs affect permeability by enhancing the passive transport across the membrane. In general, the results obtained from permeability experiments corroborate the findings observed in the apparent solubility investigation. The increase in apparent solubility had a positive impact on the passive diffusion of PTR in the GIT and BBB models. Similar results for the ASDs of polyphenol were reported by Wdowiak et al. [50] (curcumin and piperine), Sip et al. [54] (fisetin), and Deng et al. [103] (piperine).

The literature confirmed that the improved apparent solubility might positively affect the antioxidant and neuroprotective properties of the compound [23,50,60,61,70,104,105,106].

The research results show that PTR has potent antioxidant properties. Through the use of two in vitro radical scavenging tests, the ability to limit the generation and/or scavenging of free radicals was evaluated. The capacity of PTR to inhibit radical formation was measured using an ABTS assay, whereas the radical scavenging activity was measured using a DPPH assay. In our study, PTR-PVP demonstrated the ability to scavenge DPPH radicals and inhibit the formation of ABTS radicals. As a result, aqueous solutions of PTR-PVP30/PTR-PVPVA64 showed the following activities in individual tests: ABTS: IC_50_ = 52.37 ± 0.96 μg·mL^−1^ and IC_50_ = 52.99 ± 0.77 μg·mL^−1^, respectively (statistically significant difference), and DPPH: IC_50_ = 163.43 ± 7.61 μg·mL^−1^ and IC_50_ = 173.96 ± 7.90 μg·mL^−1^, respectively (statistically significant difference). This activity is related to the ability of PTR to serve as a donor of hydrogen atoms or electrons. Acharya et al. confirmed that PTR has better antioxidant activity in the ABTS test than in the DPPH test [107].

Next, the CUPRAC and FRAP assays were employed to evaluate the ability of PTR-PVP ASD to reduce metal ions. The CUPRAC assay measured the capability of PTR to reduce cupric ions (Cu^2+^) to cuprous ions (Cu^+^), while the FRAP assay was used to measure its ability to reduce ferric ions (Fe^3+^) to ferrous ions (Fe^2+^) [61,106].

In these assays, a higher absorbance indicates a greater antioxidant capacity. The absorbance of the aqueous solutions of PTR-PVP30/PTR-PVPVA64 were as follows: CUPRAC: IC_0.5_ = 122.27 ± 7.67 μg·mL^−1^/IC_0.5_ = 129.59 ± 8.92 μg·mL^−1^ (statistically significant difference), and FRAP: IC_0.5_ = 95.69 ± 9.81 μg·mL^−1^/IC_0.5_ = 98.57 ± 8.82μg·mL^−1^ (statistically significant difference). The results demonstrated the ability of a PTR to reduce metal ions in a concentration-dependent manner (higher concentration of PTR = higher activity). A positive outcome in these assays implies a higher capacity to counteract oxidative stress, which could potentially slow down or prevent the onset and progression of neurodegenerative diseases. In recent years, CUPRAC and FRAP assays have been widely used to assess the antioxidant activity of various compounds, including those with potential importance for neurodegenerative diseases. These assays have been instrumental in evaluating the antioxidant potential of fisetin [54], greek sage [108], naringenin [104], and *Cannabis sativa* extracts [109].

The regulation of cholinergic neurotransmission in the human central nervous system is largely dependent on the enzymes acetylcholinesterase (AChE) and butyrylcholinesterase (BChE). Acetylcholine is a neurotransmitter that is necessary for memory, cognitive function, and muscular control. In Alzheimer’s and Parkinson’s diseases, the inhibition of AChE and BChE can raise the concentration of acetylcholine in the synaptic cleft, enhancing neurotransmission and cognitive function [110]. The reduction in amyloid- deposition, enhancement of neurotrophic factor expression, and modulation of inflammatory responses are assumed to be the mechanisms by which AChE and BChE inhibition exert their anti-neurodegenerative actions [111].

In our study, to gain insight into the intermolecular interactions, molecular docking studies were performed for PTR at the active site 3D space of both the AChE and BChE using the Autodock Tool 1.5.7 (ADT). ADT is a widely used software tool for conducting molecular docking simulations. It allows the user to predict how a small molecule binds to a protein’s active site by evaluating various binding conformations and estimating binding energies [47,105].

The active site of the AChE enzyme was determined to interact with PTR through residues TYR-68, ASP-70, TRP-82, TYR-120, and TYR-120. PTR binds through two hydrogen bonds with ASP-70 and TYR-68, four hydrophobic interactions with ASP-70, TRP-82, and TYR-120, and two π-stacking interactions with TRP-82 and TYR-333 (Figure 7). The estimated free energy of binding (G) and estimated inhibition constant (K_i_) for the ‘PTR–AChE interaction’ were determined to be −9.98 kcal·mol^−1^ and 48.60 nM, respectively. For comparison, galantamine (GAL) and donepezil (DON) were used as positive controls (FDA-approved drug for anti-AChE [112,113,114,115]), and a water molecule as a negative control. The interactions detected between the AChE and GAL were visualized in Appendix A and with DON in Appendix A. GAL (G= −10.80 kcal·mol^−1^, K_i_ = 12.16 nM) formed a hydrogen bond with TYR-120 and hydrophobic interactions with PHE-293, PHE-334, TRP-282, and TYR-337. DON (G = −10.72 kcal·mol^−1^, K_i_ = 13.82 nM) a formed hydrogen bond with PHE-291 and hydrophobic interactions with ASP-70, LEU-72, PHE-291, TYR-68, TYR-120, TYR-337, and TRP-282. ASP-70 is responsible for the strongest protein–ligand interaction as it forms a salt bridge with DON. In contrast, the water molecule does not interact with AChE.

The active site of BChE enzyme was determined to interact with PTR through residues ASN-65, ASP-67, TRP-79, GLY-112, and ILE-437. PTR binds through three hydrogen bonds with ASN-65, ASP-67, and GLY-112 and four hydrophobic interactions with TRP-79 and ILE-437 (Figure 8).

The estimated free energy of binding and estimated inhibition constant (K_i_) for the ‘PTR–BChE interaction’ were determined to be −9.44 kcal·mol^−1^ and 119.88 nM, respectively. As with AChE, GAL and DON were employed as positive controls (FDA-approved drugs for anti-BChE [112,113,114,115]), while a water molecule served as the negative control. The interactions observed between BChE and GAL are illustrated in Appendix A, while those with DON are depicted in Appendix A. GAL (G = −9.67 kcal·mol^−1^, K_i_ = 81.58 nM) formed hydrogen bonds with ALA-196, GLU-194, and GLY-113, as well as hydrophobic interactions with TRP-79 and PHE-326, and π-stacking with HIS-433. The second positive control (DON, G = −11.64 kcal·mol^−1^, K_i_ = 2.94 nM) formed hydrogen bonds with THR-117 and TYR-125 and hydrophobic interactions with ASP-67, THR-117, TRP-79, TRP-425, and TYR-435. The strongest protein–ligand interaction was the salt bridge formed between ASP-67 and DON, and the water molecule did not interact with BChE.

The inhibition constant value (K_i_) is the half-maximum inhibition of enzyme activity by a chemical compound. It is used to estimate the potential of substrate/inhibitor in enhancing/inhibiting the biological function of enzymes. Potent inhibitors are substances that have an inhibition constant of less than 100 mM, whereas non-potent inhibitors are substances with an inhibition constant of more than 100 mM [116,117,118]. Based on this, PTR and two positive controls (GAL and DON) could be potentially potent inhibitors of AChE (K_i,PTR_ = 48.60 nM, K_i,GAL_ = 12.16 nM, K_i,DON_ = 13.82 nM) and BChE (K_i,PTR_ = 119.88 nM, K_i, GAL_ = 81.58 nM, K_i,DON_ = 2.94 nM) enzymes. The obtained results show that PTR exhibits weaker inhibition of both AChE and BChE compared to GAL and DON.

In our study, in vitro AChE and BChE inhibition was assessed for pure PTR (water solubility 4.0 ± 0.2 μg·mL^−1^). No inhibition of either enzyme was observed at this concentration. However, PTR in PTR-PVP30/PTR-PVPVA64 (water solubility 410.8 ± 1.3 μg·mL^−1^ and 383.2 ± 1.0 μg·mL^−1^, respectively) inhibited the AChE enzyme at 39.1 ± 1.6%/36.2 ± 1.0% (difference was not statistically significant). The combination of PTR with polymers significantly elevated the inhibition of BChE. The concentration obtained using PTR-PVP30/PTR-PVPVA64 inhibited BChE at 76.9 ± 1.0% for PTR-PVP30, while the PVPVA64 reached 73.2 ± 4.9% (the difference was not statistically significant). For the PTR-PVP ASDs and GAL, the dose-response curves to derive the IC_50_ values in the AChE and BChE inhibition assays are presented in Appendix A, respectively. The calculated IC_50_ values are summarized in Appendix A. Based on this, PTR-PVP ASDs exhibited weaker inhibition of AChE compared to GAL and DON, whereas they demonstrated stronger inhibition of BChE than the positive controls. Differences in inhibition activity can be related to the selectivity index (SI), which was calculated based on the IC_50_ value of the tested compounds (see Appendix A). The selectivity profile of PTR had a higher selectivity for BChE over AChE (SI_PTR-PVP30_ = 1.97 and SI_PTR-PVPVA64_ = 2.01), whereas GAL and DON had a higher selectivity for AChE over BChE (SI_GAL_ = 5.70, SI_DON_ = 1104.47).

The experimental results are consistent with theoretical predictions, which, based on the K_i_ value, indicated that PTR could be a potentially potent inhibitor of AChE and BChE enzymes. The improvement of PTR solubility after its introduction into ASD translated into a change in its biological properties, in this case, the neuroprotective potential (confirmed through AChE and BChE tests). This is consistent with the literature, which confirms that improved apparent solubility can positively affect the biological properties of a compound [23,50,60,61,70,104,105,106].

## 3. Materials and Methods

### 3.1. Materials

Pterostilbene (PTR, purity > 98%) was purchased from Xi’an Tian Guangyuan Biotech Co., Ltd. (Xi’an, China). Polyvinylpyrrolidone (PVP, PVP K30, and PVP VA64), as well as Kollidon^®^ 30 and Kollidon^®^ VA64, were supplied from BASF Pharma (Burgbernheim, Germany). Hydrochloric acid, dimethyl sulfoxide, sodium chloride, and potassium dihydrogen phosphate were obtained from Avantor Performance Materials (Gliwice, Poland). We bought 98–100% formic acid from POCH (Gliwice, Poland). High-quality clean water was produced using a Direct-Q 3 UV purification system (Millipore, Molsheim, France; model Exil SA 67120). All other chemicals were from the Sigma–Aldrich Chemical Co. (Taufkirchen, Germany).

### 3.2. Preparation of Physical Mixtures and Amorphous Solid Dispersion (ASD)

Physical mixtures (ph. m.) were obtained by grinding in an agate mortar and thoroughly mixing carefully weighed (percentage ratio 20:80, 500 mg) amounts of PTR and PVP. ASDs were prepared using the ball milling method at room temperature. First, a 50 mL stainless steel jar that was made to fit the MIXER MILL MM 400 (Retsch, Haan, Germany) was filled with physical mixes with a 20% PTR content and three stainless steel balls (ϕ 12 mm). Next, the milling time and frequency were set to 30 min and 30 Hz, respectively.

The obtained ASD looked like a uniform, fine powder. The powders were stored in a desiccator for further investigation.

### 3.3. X-ray Powder Diffraction (XRPD)

Using a Bruker D2 Phaser diffractometer (Bruker, Germany), the physical state of PTR was verified for the following samples: the pure samples, the physical mixtures, and the resultant ASDs. CuKα radiation (1.54060 Å) at tube voltages of 30 kV and tube currents of 10 mA was used to record the diffraction patterns. The angular range was from 5° to 40° 2Θ with a step size of 0.02° 2Θ and a counting rate of 2 s·step^−1^. Origin 2021b software (OriginLab Corporation, Northampton, MA, USA) was used to evaluate the acquired data.

### 3.4. Thermal Analysis of Amorphous Solid Dispersions

#### 3.4.1. Thermogravimetric Analysis (TG)

PTR, PVP K30, PVP VA64, and amorphous solid dispersions were examined for thermal stability using a TG 209 F3 Tarsus^®^ micro-thermobalance (Netzsch, Selb, Germany). An 85 µL open Al_2_O_3_ crucible was filled with 7–11 mg of the sample powder. The temperature range for TG measurements was from 25 °C to 400 °C, at a constant heating rate of 10 °C per minute in a nitrogen atmosphere (flow rate 250 mL·min^−1^). Once the TG data had been gathered, it was evaluated using Proteus 8.0 (Netzsch, Selb, Germany). The outcomes were shown using a program Origin 2021b (OriginLab Corporation, Northampton, MA, USA).

#### 3.4.2. Differential Scanning Calorimetry (DSC)

The DSC experiments were carried out using a differential scanning calorimeter, model DSC 214 Polyma (Netzsch, Selb, Germany). The reference sample was a blank, sealed aluminum DSC pan with a cover, and powdered samples weighing 3.9–10.0 mg were placed into sealed pans with holes in the lid.

Using a single heating mode and a scanning rate of 10 °C per minute, the melting point of PTR in the neat compound was investigated. Melting and cooling modes were used to observe the glass transition (T_g_) of PTR, PVP K30, and PVP VA64. The nitrogen atmosphere’s flow rate was set at 250 mL per minute. The collected DSC data were examined using Proteus 8.0 (Netzsch, Selb, Germany), whereas data visualization was performed in Origin 2021b (OriginLab Corporation, Northampton, MA, USA).

The prediction of the glass transition values (Tg) of amorphous PTR-PVP ASD was determined according to our previous protocol [23]. Briefly, the mathematical models used were the Gordon–Taylor equation and the Couchman–Karash equation. The densities of PTR (1.2597 ± 0.0007 g·cm^−3^), PVP K30 (1.2119 ± 0.0129 g·cm^−3^), and PVP VA64 (1.2699 ± 0.0025 g·cm^−3^) were measured with a helium gas pycnometer (Accupyc 1340, Micrometrics Instrument Corporation, Norcross, GA, USA).

### 3.5. ATR-FTIR Spectroscopy

ATR-FTIR spectra in the MIR region (400–4000 cm^−1^) were obtained using an IRTracer-100 spectrophotometer with a QATR that holds a diamond ATR system (Shimadzu, Kyoto, Japan). The resolution was 4 cm^−1^, while 100 scans over the selected wavenumber range were averaged for each sample. All infrared spectra were collected using LabSolution IR software (version 1.86 SP2, Shimadzu, Kyoto, Japan).

### 3.6. Studies of PTR Properties after Introduction to ASD

#### 3.6.1. Physical Stability

Powder samples of amorphous solid dispersions were placed into 2 mL Eppendorf and then placed in the laboratory incubator CLN 32 (Pol-eko Aparatura, Wodzisław Śląski, Poland). Physical stability was monitored in different humidity and temperature conditions (30 °C/65% RH and 40 °C/75% RH). Due to the change in the physical form of samples stored at 40 °C/75% RH, further tests were carried out only at 30 °C/65% RH for up to 3 months. XRPD was utilized to identify the solid-state structures of the samples following a storage period of 1, 2, and 3 months.

#### 3.6.2. HPLC Analysis

All HPLC tests were carried out on a Shimadzu Nexera (Shimadzu Corp., Kyoto, Japan) [23]. A Dr. Maisch ReproSil-Pur Basic-C18 100 column with 5 µm and 100 × 4.60 mm particle sizes (Dr. Maisch, Ammerbuch-Entringen, Germany) was utilized for the stationary phase. Methanol/0.1% formic acid (70:30 *v*/*v*) was used as the mobile phase. The mobile phase was vacuum-filtered via a 0.45 µm nylon filter. The experimental parameters were as follows: flow rate of 1.0 mL·min^−1^, wavelength of 308 nm, and column temperature of 35 °C.

#### 3.6.3. Apparent Solubility

In this experiment, 40 mg of PTR and 250 mg of ASDs were placed into glass vials together with 4 mL of distilled water. All samples were mixed using a vortex mixer for 30 s. The suspensions were then filtered via a 0.22 µm filter and ran through an HPLC analysis procedure. The analysis was performed in triplicate.

#### 3.6.4. Dissolution Rate Studies

This study used the method proposed by Mesallati et al. [119] with modifications. The test was carried out at a temperature of 37 °C in 50 mL beakers containing 30 mL of pH 6.8 buffer. An excess amount of sample was added to the beakers, which were stirred at 250 rpm. Samples were drawn from the vial at specific time points over 7 h. The aliquots were filtered with a 0.45 µm membrane filter. The amount of PTR was monitored using HPLC Shimadzu Nexera (Shimadzu Corp., Kyoto, Japan) with the previously described method (Section 3.6.2. HPLC Analysis).

#### 3.6.5. Permeability Studies

Based on a previous report [50], in vitro gastrointestinal (GIT) and blood–brain barrier (BBB) permeability were studied using the Parallel Artificial Membrane Permeability Assay (PAMPA) models. The HPLC method was employed for the examination of PTR concentrations, and the subsequent formulas were utilized for the computation of the apparent permeability coefficient (P_app_) [78]:(2)$$P_{app}=\frac{-\ln\left(1-\frac{C_{A}}{C_{equilibrium}}\right)}{S\times \left(\frac{1}{V_{D}}+\frac{1}{V_{A}}\right)\times t}$$
where V_D_—donor volume, V_A_—acceptor volume, C_equilibrium_—equilibrium concentration $C_{equilibrium}=\frac{C_{D}\times V_{D}+C_{A}\times V_{A}}{V_{D}+V_{A}}$, C_D_—donor concentration, C_A_—acceptor concentration, S—membrane area, and t—incubation time (in seconds).

According to the literature [78,120], substances categorized as having low permeability typically exhibit a Papp value below 0.1 × 10^−6^ cm·s^−1^. Compounds designated as moderately permeable fall within the range of 0.1 × 10^−6^ cm·s^−1^ ≤ Papp < 1 × 10^−6^ cm·s^−1^, while those with a Papp value ≥ 1 × 10^−6^ cm·s^−1^ are classified as highly permeable.

#### 3.6.6. Antioxidant Activity

The antioxidant activity of the PTR, PTR-PVP30 ASD, and PVPVA64 ASD water solutions was assessed using four different assays: ABTS (2,2′-Azino-bis(3-ethylbenzthiazoline-6-sulfonic acid), DPPH (2,2-Diphenyl-1-picrylhydrazyl), CUPRAC (cupric-reducing antioxidant capacity), and FRAP (ferric-reducing antioxidant power). All assays were carried out in a 96-well plate, and the samples were measured using spectrophotometry.

The ABTS assay was carried out in accordance with the previously reported procedure [121]. First, 25.0 µL of each sample was pipetted to 175.0 µL of the ABTS solution (blank was a mixture of 175.0 µL ABTS solution and water). Incubation was performed in darkness at room temperature for 30 min while shaking. The absorbances were read at 734 nm on a plate reader (Multiskan GO, Thermo Fisher Scientific, Waltham, MA, USA).

For the DPPH assay, the procedure outlined by Stasiłowicz-Krzemień et al. [122] was followed. First, 25.0 µL of each sample was pipetted to 175.0 µL of the DPPH solution (blank was a mixture of 175.0 µL DPPH solution and water). Incubation was performed in darkness at room temperature for 30 min while shaking. The absorbances were read at 517 nm on a plate reader (Multiskan GO, Thermo Fisher Scientific, Waltham, MA, USA).

The percentage of the inhibition of ABTS and DPPH radicals by the samples was determined using Equation (3).
(3)$$\mathrm{The}\;\mathrm{degree}\;\mathrm{of}\;\mathrm{radical}\;\mathrm{scavenging}\;(\%)=\frac{A_{0}-A_{i}}{A_{0}}\cdot 100\%,$$
where A_0_ is the absorbance of the control and A_i_ is the absorbance of the sample. Each measurement was performed six times.

The CUPRAC and FRAP assays were used to evaluate the reducing potential of samples. For the CUPRAC assay, following the Gościniak et al. method [123], an amount of 50.0 µL of each sample solution and 150.0 µL of the CUPRAC reagent was pipetted onto the plate and incubated for 30 min at room temperature in darkness. For the FRAP assay, following the Sip et al. method [124], the mixture of 25.0 µL of each sample solution and 175.0 µL of FRAP mixture was incubated for 30 min at 37 °C in dark conditions. Then, the absorbance was measured at 450 nm (CUPRAC assay) and 593 nm (FRAP assay) using a plate reader (Multiskan GO, Thermo Fisher Scientific, Waltham, MA, USA). The measurements were also performed in six replicates.

#### 3.6.7. Anticholinesterase Activity

A spectrometric-modified test by Ellman et al. was used to inhibit AChE and BChE [125]. Thiocholine and artificial substrates are needed for this approach. Through enzymatic interactions with 5,5′-dithio-bis-(2-nitrobenzoic) acid (DTNB), thiocholine is released, resulting in the formation of the 3-carboxy-4-nitrothiolate anion (TNB anion).

The increase in the color of the thiocholine on a 96-well plate is used as a spectrophotometric indicator of enzyme activity. The process for preparing the PTR, PTR-PVP30, and PTR-PVP VA64 water solutions for the assay is described in Section 3.6.2. The galantamine (dissolved in DMSO) was used as a reference (range of concentration 7.8–250 µg·mL^−1^). Apparent solubility was followed. The wells contained 25.0 µL of the test solution and 30.0 µL of AChE/BChE solution at a concentration of 0.2 U·mL^−1^, and 40.0 µL of 0.05 M Tris-HCl the buffer with a pH of 8.0. These were incubated at room temperature for five minutes while being shaken. The well was then filled with 125.0 µL of the 0.3 mM DTNB solution and 30.0 µL of the 1.5 mM acetylthiocholine iodide (ATCI)/butyrylthiocholine iodide (BTCI) solution and incubated for 20 min under the same conditions.
(4)$$\mathrm{AChE}/\mathrm{BChE}\;\mathrm{inhibition}\;(\%)=\frac{1-\left(A_{1}-A_{1b}\right)}{\left(A_{0}-A_{0b}\right)}\cdot 100\%$$
where A_1_ is the absorbance of the test sample; A_1b_ is the absorbance of the blank of the test sample; A_0_ is the absorbance of the control; A_0b_ is the absorbance of the blank of the control.

#### 3.6.8. Molecular Docking Study

Molecular docking studies were performed to investigate the binding mode between pterostilbene and the AChE and BChE enzymes [126]. The active site of AChE and BChE was predicted using PrankWeb (https://prankweb.cz/, accessed on 3 July 2023) [127,128,129], while the preparation of cholinesterases and the compound was performed using AutoDock Tools 1.5.7 (ADT; Scripps Research Institute, La Jolla, San Diego, CA, USA) and OpenBabel [130]. Molecular docking was carried out using AutoDock Vina 1.1.2. Structural interactions were visualized and analyzed using Protein–Ligand Interaction Profiler (PLIP server, https://plip-tool.biotec.tu-dresden.de/, accessed on 12 July 2023) [131] and PyMOL (DeLano Scientific LLC, Palo Alto, CA, USA).

The molecular structure of the PTR was downloaded from PubChem (PubChem CID: 5281727; website: https://pubchem.ncbi.nlm.nih.gov/, accessed on 2 December 2022) in sdf format. The PTR’s geometries were optimized using the Gaussview (Wallingford, CT, USA) program (B3LYP/6-31 (d,p)) before molecular docking. X-ray crystal structures of human AChE (PDB code: 4BDT with 3.10 Å resolution) and human BChE (PDB code: 4BDS with 2.10 Å resolution) were retrieved from the Protein Data Bank (PDB) (https://www.rcsb.org/, accessed on 4 February 2023) in PDB format. These receptors were prepared using AutoDock Tools.

Briefly, water molecules and bound ligands were removed, polar hydrogens and Kollman charges were added, and the non-polar hydrogens were merged. The distance between the surface area of the enzymes and the PTR molecule was limited to the maximum radius limit of 0.375 Å. The grid box was positioned around the active site pocket predicted using PrankWeb (https://prankweb.cz/, accessed on 3 July 2023) [127,128,129].

The AChE’s active pocket contained: GLY-116, GLY-117, GLY-118, TYR-120, SER-121, GLY-122, LEU-126, GLU-198, SER-199, TRP-282, LEU-285, SER-289, PHE-291, ARG-292, PHE-293, TYR-333, PHE-334, TYR-337, TRP-435, HIS-443, GLY-444, TYR-445, TYR-68, ASP-70, TYR-73, THR-79, TRP-82, and ASN-83.

The BChE’s active pocket contained: GLY-112, GLY-113, GLY-114, GLN-116, THR-117, GLY-118, TYR-125, GLU-194, SER-195, TRP-228, PRO-282, LEU-283, SER-284, VAL-285, ALA-325, PHE-326, TYR-329, PHE-393, TRP-425, HIS-433, GLY-434, TYR-435, ASP-67, GLY-75, SER-76, TRP-79, and ASN-80.

The grid box for AChE had dimensions of 85/100/105 and was centered at x = −4.5329, y = −37.0163, and z = −50.394. The BChE grid box, on the other hand, had dimensions of x = 70, y = 60, and z = 70 and was centered at x = 135.1815, y = 115.905, and z = 40.294. Using Autodock Vina, which employs the Broyden–Fletcher–Goldfarb–Shanno (BFGS) algorithm through an Iterated Local Search approach to produce several ligand conformers, it was possible to predict the interactions and binding affinity of the protein-ligand complex [132]. In terms of scoring, Autodock Vina employs a hybrid score system that combines knowledge- and empirical-based factors [132]. Finally, the results were saved in PDBQT format.

Following docking simulations, the pose with the highest score was chosen and exported to PDBQT format. The file was opened in the Protein–Ligand Interaction Profiler (PLIP server, https://plip-tool.biotec.tu-dresden.de/, accessed on 12 July 2023) after being converted to the PDB format using the Open Babel 3.1.1 program [130,131,133]. Using the PLIP server, interactions between PTR and the enzymes’ active sites were identified. The PLIP server then downloaded a file in PSE format. The PyMOL 2.5.1 tool (DeLano Scientific LLC, Palo Alto, CA, USA) was used to visualize the docked complexes of the enzyme and PTR (file in PSE format) [134].

### 3.7. Statistical Analysis

Statistical analyses were performed using Statistica 13.3 (StatSoft, Krakow, Poland). The data were analyzed using a one-way analysis of variance (ANOVA) followed by Duncan’s post-hoc test. A probability level of *p* < 0.05 was considered statistically significant. The results are reported as mean ± standard deviations.

## 4. Conclusions

Using the milling process, we successfully obtained amorphous solid dispersions (ASDs) of pterostilbene (PTR) with PVP30 and PVPVA64 polymers. FT-IR spectroscopy provided evidence confirming that hydrogen bonds are responsible for maintaining the amorphous form of PTR. Whereas, DSC analysis confirmed miscible ASDs.

Our study revealed (compared to pure PTR) enhancements in apparent solubility, a released profile in pH 6.8, in vitro permeability, antioxidant properties, and neuroprotective effects.

The obtained ASDs of PTR enabled the maintenance of the supersaturated state in the released rate study. This indicates that PVP30 and VA64 effectively inhibited the crystallization of PTR. For PTR-PVP30 and PTR-PVPVA64 ASDs, after 60 min of dissolution, the total amount of PTR was ~1417-fold and ~458-fold higher than pure PTR, respectively. Moreover, the amorphous PTR demonstrated enhanced permeability across biological membranes (in vitro GIT and BBB assays), which may potentially translate into increased bioavailability. Increasing the permeability of the blood–brain barrier (BBB) can enhance the neuroprotective properties of PTR. This heightened permeability allows for an improved delivery of the PTR to the central nervous system, potentially augmenting its efficacy in providing neuroprotection. The enhanced BBB penetration facilitates better access of the PTR to the brain, optimizing its impact on neuroprotective mechanisms and therapeutic outcomes. Understanding and manipulating BBB permeability is crucial for developing strategies to enhance the neuroprotective potential of various active pharmaceutical ingredients in the context of neurological disorders.

In addition, ASDs demonstrated stability under storage conditions (30 °C/65% RH), demonstrating their potential for long-term use in pharmaceutical production.

These findings contribute to the understanding of how PTR amorphization affects physicochemical and biological properties. Future research in this direction may uncover additional applications and opportunities for enhancing the therapeutic potential of bioactive compounds through amorphous formulations.

## Figures and Tables

**Figure 1 ijms-25-02774-f001:**
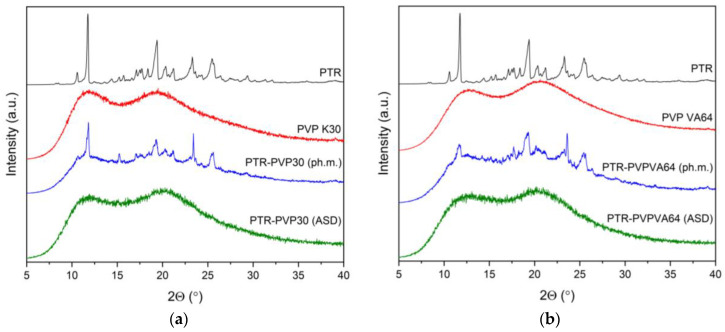
XRPD analysis, range 5–40° 2Θ: (**a**) pterostilbene (PTR, black line), PVP K30 (red line), pterostilbene-PVP30 physical mixture (PTR-PVP30 (ph.m.), blue line), pterostilbene-PVP30 amorphous solid dispersion (PTR-PVP30 (ASD), green line); (**b**) pterostilbene (PTR, black line), PVP VA64 (red line), pterostilbene-PVPVA64 physical mixture (PTR-PVPVA64 (ph.m.), blue line), pterostilbene-PVPVA64 amorphous solid dispersion (PTR-PVPVA64 (ASD), green line).

**Figure 2 ijms-25-02774-f002:**
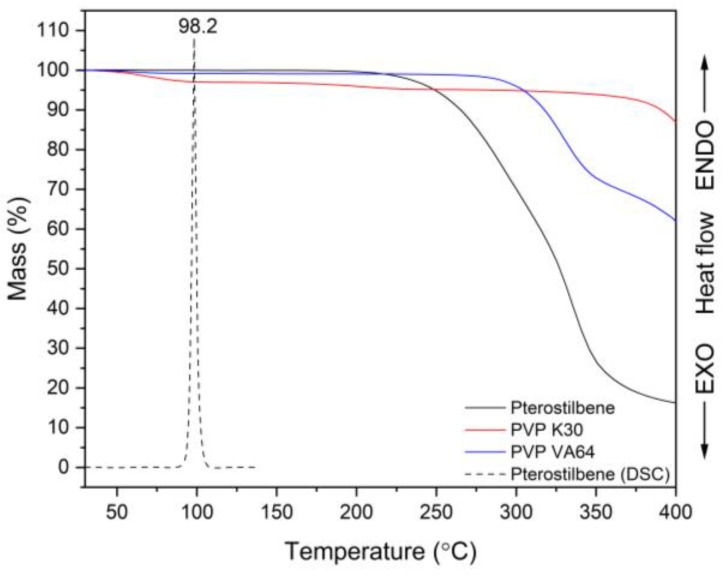
Thermal analysis: TG thermogram of pterostilbene (black line), TG thermogram of PVP K30 (red line), TG thermogram of PVP VA64 (blue line), DSC thermogram of pterostilbene (black dashed line, first heating).

**Figure 3 ijms-25-02774-f003:**
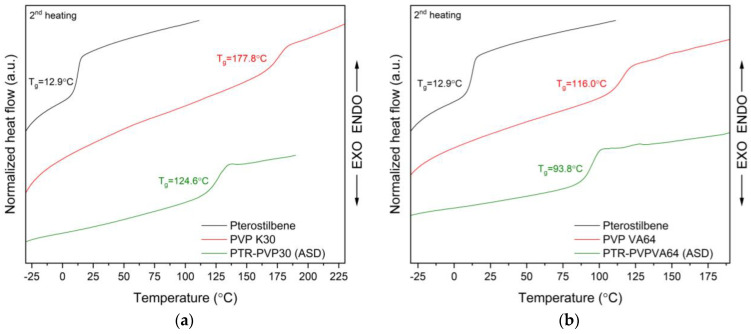
DSC analysis, second heating: (**a**) amorphous pterostilbene (black line), PVP K30 (red line), PTR-PVP30 amorphous solid dispersion (PTR-PVP30 (ASD), green line); (**b**) amorphous pterostilbene (black line), PVP VA64 (red line), PTR-PVPVA64 amorphous solid dispersion (PTR-PVPVA64 (ASD), green line).

**Figure 4 ijms-25-02774-f004:**
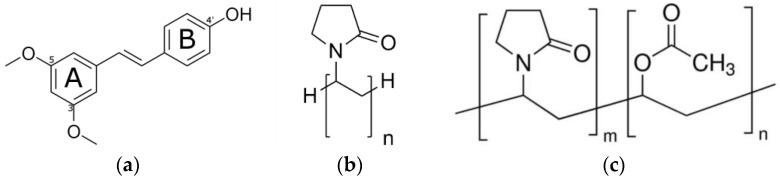
Structure of (**a**) pterostilbene; (**b**) PVP K30; (**c**) PVP VA64. Legend: rings A and B—two phenolic rings of pterostilbene.

**Figure 5 ijms-25-02774-f005:**
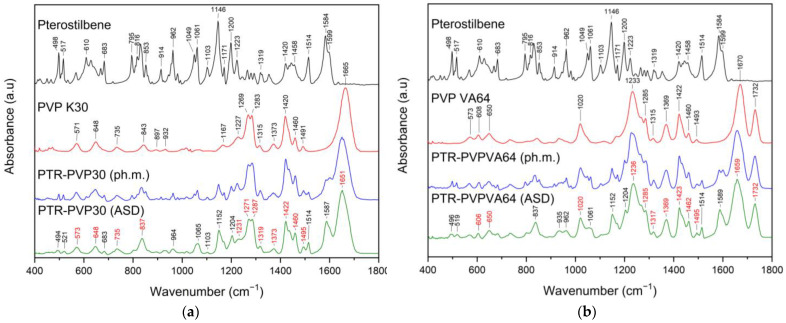
FTIR-ATR analysis, range 400–1800 cm ^−1^: (**a**) pterostilbene (black line), PVP K30 (red line), pterostilbene-PVP30 physical mixture (PTR-PVP30 (ph.m.), blue line), pterostilbene-PVP30 amorphous solid dispersion (PTR-PVP30 (ASD), green line); (**b**) pterostilbene (black line), PVP VA64 (red line), pterostilbene-PVPVA64 physical mixture (PTR-PVPVA64 (ph.m.), blue line), pterostilbene-PVP VA64 amorphous solid dispersion (PTR-PVPVA64 (ASD), green line). Red-colored numbers correspond to the PVP bands in ASDs.

**Figure 6 ijms-25-02774-f006:**
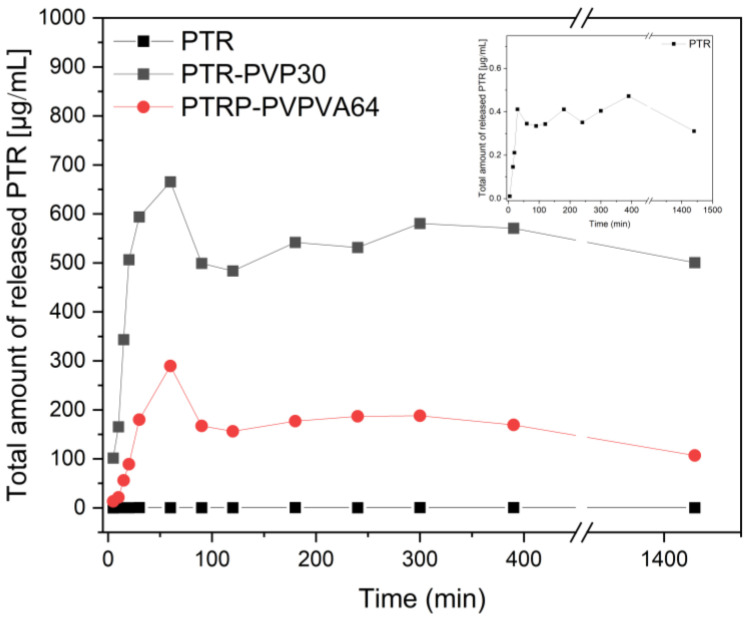
The released profiles of pterostilbene (PTR) and PTR-PVP amorphous solid dispersions (PTR-PVP30 and PVP-VA64) at pH 6.8.

**Figure 7 ijms-25-02774-f007:**
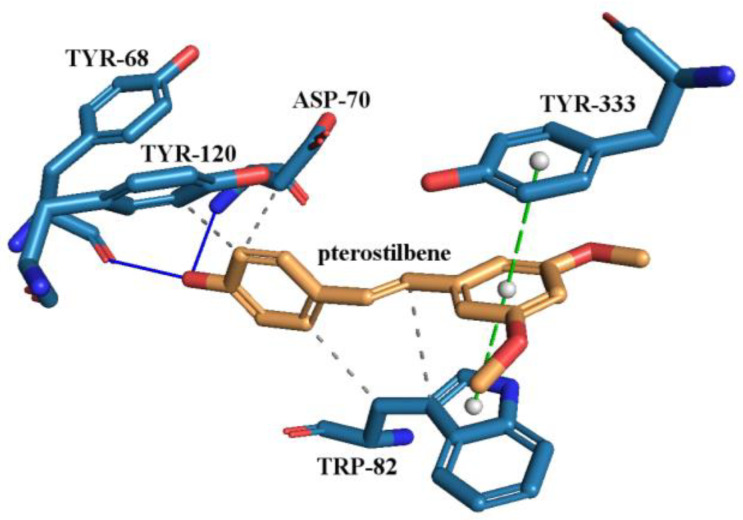
Proposed binding mode of pterostilbene (PTR) with human acetylcholinesterase (AChE, PDB id: 4BDT). The key interactions of PTR with residues in the active sites of AChE. Legend: ASP—aspartic acid, TRP—tryptophan, TYR—tyrosine, grey dashed line—hydrophobic interaction, blue solid line—hydrogen bond, green dashed line—π-stacking.

**Figure 8 ijms-25-02774-f008:**
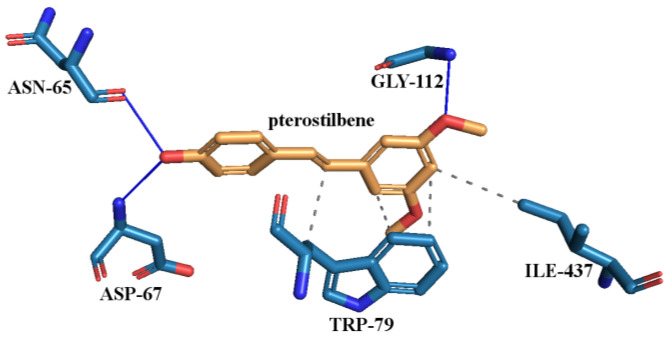
Proposed binding mode of pterostilbene (PTR) with human butyrylcholinesterase (BChE, PDB id: 4BDS). The key interactions of PTR with residues in the active sites of BChE. Legend: ASN—asparagine, ASP—aspartic acid, GLY—glycine, ILE—isoleucine, TRP—tryptophan, grey dashed line—hydrophobic interaction, blue solid line—hydrogen bond.

**Table 1 ijms-25-02774-t001:** Summary of the most important parameters of thermal analysis of PTR-PVP amorphous solid dispersions (PTR-PVP ASDs) and experimental and theoretical T_g_ values.

	Mass (mg)	ΔC_p_ (J·(g·°C)^−1^)	T_g,exp_ (°C)	T_g,G-T_ (°C)	T_g,C-K_ (°C)	Deviation
PTR	7.20	0.461	12.9	-	-	
PVP K30	7.58	0.154	177.8	-	-	
PVP VA64	8.40	0.311	115.3	-	-	
PTR-PVP30 ASD	7.50	0.329	124.6	51.1	107.2	+
PTR-PVPVA64 ASD	7.08	0.404	93.8	44.4	87.6	+

PTR—pterostilbene, PVP—polyvinylpyrrolidone, PTR-PVP ASD—PTR-PVP amorphous solid dispersions, ΔC_p_—heat capacity, T_g,exp_—glass transition temperature (experimental), T_g,G-T_—glass transition temperature (calculated by Gordon–Taylor equation), T_g,C-K_—glass transition temperature (calculated by Couchman–Karasz equation), +—positive deviation.

## Data Availability

The data are contained within the article and Appendix A.

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
