# Peer review of "Enhanced Antioxidant and Neuroprotective Properties of Pterostilbene (Resveratrol Derivative) in Amorphous Solid Dispersions"

_ijms, 2024, doi:10.3390/ijms25052774_

Round 1
Reviewer 1 Report
Comments and Suggestions for Authors
1-The title, the context in Introduction and the aim of paper should be better aligned with the aim of Journal.
2-Positive and negative controls should be used in the docking experiment.
3-The date of access of all online softwares should be included.
4- In the Introduction section the author should refer the following research paper and comment on recent in-silico techniques. It will be good information for the readers.
Therapeutic Potential of Myrrh, a Natural Resin, in Health Management through Modulation of Oxidative Stress, Inflammation, and Advanced Glycation End Products Formation Using In Vitro and In Silico Analysis. https://doi.org/10.3390/app12189175.
5- Quality of various figures is very poor. Improve it.
6-The novelty character of paper should be better marked.
7- Provide the reference of various methodologies in material method section.
8- The conclusion should have achievements, shortcomings, and prospects for future developme11- Careful attention needs to be paid to correct all the grammatical errors and reduce the repetition of the same explanations in different sections of the manuscript.nt etc. I think that conclusion should be expanded.
9- The authors are advised to highlight the efficacy and bioavailability of pterostilbene in amorphous solid dispersions.
10- To give a strong argument for why pterostilbene would be a superior therapeutic choice in this aspect than stilbene, I propose including antioxidant and neuroprotective clinical data for stilbene.
Comments on the Quality of English Language11- Careful attention needs to be paid to correct all the grammatical errors and reduce the repetition of the same explanations in different sections of the manuscript.
Reviewer 2 Report
Comments and Suggestions for Authors
The manuscript titled as “Enhanced Antioxidant and Neuroprotective Properties of 2 Pterostilbene in Amorphous Solid Dispersions” with manuscript ID IJMS-2845672 has been submitted to International Journal of Molecular Sciences (IJMS) for possible publication is not suitable for the publication and should be Majorily Revised due to following points.
1. Introduction should include some information on amorphous solid dispersion.
2. If the drug have bioavailability and solubility issue than in introduction part author mentioned Obrador et al drug is safe when administered intravenously. so the bioavailability of drug can be achieved through IV formulation and the solubility of drug generally is enhanced by reducing size and by temperature increases why dispersion solid medium is used.
3. No in vivo studies are performed to confirm the neuroprotective properties of drug that the author claim in his research.
4. Anticholinergic activity and molecular docking study in method will be easy to understand if headings are separated.
5. FRAP and CUPRAC results should be more elaborated.
6. No references in methods were mentioned. Why? All methods are developed by the authros.
7. What was original physical state of PTR ASD? (In stability studies, you discussed previous state of ASD)
8. The manuscript needs severe improvements in language and proofreading. For example following sentence is meaningless.
The analysis was performed in triplicate.4. Conclusions
This section is not mandatory but can be added to the manuscript if the discussion is unusually long or complex
Comments on the Quality of English LanguageLanguage should be improved
Reviewer 3 Report
Comments and Suggestions for Authors
Authors have performed very exhaustive in vitro characterization. Some queries are
1." Solid dispersion enhanced the solubility " is a well known fact. what is the need of it?
2. Authors must have compared all methods of preparation of solid dispersion and must have reported the best mehod for Pterostilbene.
3. What is stability of prepared solid dispersion.
4. It would be great if authors could perform in vivo study to report Bioavailability enhancement.
5. A similar paper was already published by the same authors in 2023 February in "Pharmaceutics"
6. Plagiarism was checked by Turnitin and it was found 35%
Reviewer 4 Report
Comments and Suggestions for Authors
1. The donepezil is FDA approved drug for anti-AChE, therefore, the pose of donepezil-AChE docking should be applied to pterostilbene-AChE in the present study. The key amino acid residues in the peripheral anionic site and catalytic anionic site should be re-examined. Please check the references of (a) Synthesis, kinetic evaluation and molecular docking studies of donepezil-based acetylcholinesterase inhibitors. J. Mol. Structure 2022, 1247, 131425; (b) Identification of novel acetylcholinesterase inhibitors designed by pharmacophore-based virtual screening, molecular docking and bioassay. Sci. Reports 2018, 8, 14921;(c) Inhibition of acetylcholinesterase and amyloid-β aggregation by piceatannol and analogues: asessing in vitro and in vivo impact on a murine model of scopolamine-induced memory impairment. Antioxidants 2023, 12, 1362.
2. The free pterostilbene releases from amorphous solid dispersions in the simulated gastric fluids or intestinal fluids (as the physiological conditions) determined by HPLC should be added in the revised MS.
3. Please showed the results of dose-effects of anti-AChE activities of pterostilbene and donepezil and calculated the IC50 in the revised MS. The nanoM scale of IC50 should provide the data.
Round 2
Reviewer 2 Report
Comments and Suggestions for Authors
The manuscript titled as “Enhanced Antioxidant and Neuroprotective Properties of 2 Pterostilbene in Amorphous Solid Dispersions” with manuscript ID IJMS-2845672-V2 has been submitted to International Journal of Molecular Sciences (IJMS) for possible publication is now suitable for the publication in IJMS as such.
1. Ensure that all abbreviations have been completely explained when firstly used.
2. Try to improve the Diagrams quality.
Comments on the Quality of English LanguageSeems ok now
Reviewer 4 Report
Comments and Suggestions for Authors
Please showed the effects of concentrations (three or four doses) on AChE activities of pterostilbene and donepezil in vitro rather than the molecular dockings in the revised MS.
